# Dietary Pattern Associated with the Risk of Poor Glycemic Control in Chinese Diabetic Adults: Results from the China Nutrition and Health Surveillance 2015–2017 Survey

**DOI:** 10.3390/nu15010056

**Published:** 2022-12-23

**Authors:** Yuqian Li, Lianlong Yu, Zhen Liu, Shanshan Jia, Qingqing Man, Qianrang Zhu, Cheng Li, Yuxiang Yang, Beibei Liu, Jian Zhang

**Affiliations:** 1National Institute for Nutrition and Health, Chinese Center for Disease Control and Prevention, Beijing 100050, China; 2Shandong Center for Disease Control and Prevention, Jinan 250014, China; 3NHC Key Laboratory of Trace Element Nutrition, National Institute for Nutrition and Health, Chinese Center for Disease Control and Prevention, Beijing 100050, China; 4Jiangsu Provincial Center for Disease Control and Prevention, Nanjing 210009, China; 5Department of Clinical Nutrition, Beijing Friendship Hospital, Capital Medical University, Beijing 100050, China

**Keywords:** dietary pattern, glycemic control, surveillance, reduced rank regression, macronutrients intake

## Abstract

The present study was conducted to investigate the nutrition status of dietary patterns and the association between dietary patterns and the risk of poor glycemic control in Chinese diabetics. The relevant data was collected from the China Nutrition and Health Surveillance 2015–2017 survey, which is a national cross-sectional surveillance program. A total of 2031 participants were included in the present statistical analysis. Food consumption was assessed through a validated and standard food frequency questionnaire. Dietary patterns were derived with reduced rank regression using hemoglobin A1c. Diabetes was diagnosed by medical institutions, glycemic control was defined as hemoglobin A1c less than 7%, poor glycemic control was defined as hemoglobin A1c greater than 7%. A multiple-variable-adjusted logistic regression, including age, living area, income level, educational attainment, body mass index, occupational physical activity, energy intake, current smoking status, current drinking status, diabetic medication use, insulin use, following diabetic diets, increased exercise, and glucose monitoring, was adjusted to explore the association between dietary patterns and the risk of poor glycemic control in diabetes. Two gender-specific dietary patterns have an increased risk of poor glycemic control and are characterized by a low intake of freshwater fish, poultry, and fruits. For male participants, the dietary pattern was characterized by a high intake of wheat and its products, a low intake of vegetables, corn and its products, shrimp and crab, and beans (Q4 vs. Q1, *OR =* 2.69, 95% *CI*: 1.76 to 4.10). For female participants, the dietary pattern was characterized by a low intake of snacks and nuts, and algae and mushroom (Q4 vs. Q1, *OR* = 2.18, 95% *CI*: 1.48 to 3.20).

## 1. Introduction

Diabetes mellitus is one of the most common chronic diseases in the world, characterized by pancreatic β-cell dysfunction and peripheral insulin resistance. The clinical manifestations are elevated hyperglycemia or/and impaired glucose tolerance. Around 90% of people with diabetes have type 2 diabetes, associated with the risk factors of family history of diabetes, physical activity, poor nutrition during pregnancy, unhealthy diet, and so on. Worldwide, 537 million adults (20–79 years) are living with diabetes. This number is predicted to rise to 643 million by 2030 and 783 million by 2045 [1]. In mainland China, the latest study in this area included 173,642 participants in 2018, the estimated overall prevalence of diabetes was 12.4%, and the prevalence of diabetes had risen 1.5% compared with that in 2013 [2].

The key to diabetes management is glycemic management, and the aim of glycemic management (lowering the blood glucose level) is to reduce the complications of diabetes. For diabetics, glycemic management is primarily assessed with the hemoglobin A1C (HbA_1c_; %) test, a reasonable HbA_1c_ target for most non-pregnant adults with sufficient life expectancy to see microvascular benefits is around 7% or less [3]. Bad glycemic management yields substantial and enduring growth in the onset and progression of microvascular complications, such as retinopathy, diabetic nephropathy, and peripheral neuropathy. In a population-based cohort study, compared with diabetes with an HbA_1c_ of 6.5 to 7.0%, diabetes with an HbA_1c_ of 10.0% or higher had increased COVID-19 related mortality (hazard ratio (*HR*) = 1.61, 95% confidence interval (*CI*), 1.47 to 1.77, *p* < 0.0001) in type 2 diabetes [4]. Therefore, during the period of the global epidemic of COVID-19, glycemic management in diabetic patients is important.

In a meta-analysis, including 24 studies with 369,251 people, glycemic control was 42.8% (95% *CI*: 38.1% to 47.5%) in 20 countries from 2006 to 2012 [5]. In China, the weighted prevalence of glycemic control in Chinese diabetics was 50.1% (95% *CI*: 47.5% to 52.6%) in 2018 [2]. However, there is still a gap compared to the goal of a 60% glycemic control rate in Chinese diabetics by 2015 as proposed in the *China Chronic Disease Prevention and Control Work Plan (2012–2015)* [6]. Public health institutions need to make more efforts to improve the glycemic control of diabetic patients; dietary pattern (DP) may make a difference.

Healthy DPs, characterized by the increased consumption of vegetables, fruits, legumes, nuts, whole grains, unsaturated vegetable oils, fish, and lean meat or poultry (when meat was included) and a low intake of red and processed meat, high-fat dairy, and refined carbohydrates or sweets, are associated with a decreased risk of all-cause mortality [7]. DPs were also recommended as medical nutrition therapy (MNT) in the dietary guidelines of the American Diabetes Association (ADA) and the European Association for the Study of Diabetes (EASD) [8]. DPs such as low carbohydrate diets, dietary approaches to prevent hypertension (DASH), vegan and vegetarian diets, a Mediterranean diet, a Ma-Pi macrobiotic diet, an intermittent fasting diet, a low glycemic index diet, a high protein diet, and a high fiber diet all improve the glucose metabolism, including a reduction in fasting plasma glucose, HbA_1c_, and insulin resistance [9,10,11,12,13,14,15,16,17,18]. Among the dietary patterns mentioned above, the effect of a Mediterranean diet appears to be the greatest [8,19,20]. However, the Mediterranean diet may not be appropriate everywhere; Chinese diabetic patients should be advised to prefer healthy food groups in local lists of foods.

Principal component analysis (PCA), factor analysis (FA), and reduced rank regression (RRR) were widely used in exploring the DP in nutritional epidemiology. RRR can be used by choosing disease-specific response variables and determining the linear functions of predictors (foods) by maximizing the explained variation in responses (disease-related nutrients or food groups) [21]. Compared with PCA and FA, RRR has the advantage of assessing the association between DPs and the development of diseases by combining prior information and dietary information.

The data from the China Nutrition and Health Surveillance 2015–2017 were analyzed in the present study to explore the association between DP derived from RRR and the risk of poor glycemic control among Chinese diabetic patients.

## 2. Materials and Methods

### 2.1. Participants

The cross-sectional data were part of the China Nutrition and Health Surveillance 2015–2017 (CNHS 2015–2017). This survey was conducted among adults aged 18 years and older in 2015 in mainland China. The participants were enrolled in 2015. The sampling design was based on a stratified, multistage, and random sampling method to extract the representative samples from 31 provinces/municipalities/autonomous districts in mainland China in 2015. After four stages of stratified sampling, in each villager/resident group, among the 45 households that were finally determined to participate in the survey, 20 households were selected to enroll in the dietary survey [22]. The inclusion criteria were as follows: (1) participants aged 18 years and older; (2) participants had completed all the parts of the survey, including the basic information interview, dietary survey, anthropometric measurement, and laboratory test; (3) individuals with an average energy intake, which is 800–4800 Kcal per day for males and 500–4000 Kcal per day for females, calculated by the Food Frequency Questionnaire (FFQ); and (4) individuals diagnosed with diabetes by medical institutions. We finally analyzed 2031 participants in the current study. All the participants signed informed consent forms at the beginning of the survey, which was supported by the Ethics Committee of the Chinese Center for Disease Control and Prevention (approval number: 201519-B).

### 2.2. Basic Information Collection

A standard questionnaire was used to collect information on the social-demographic characteristics of participants. The general information questionnaires were carried out by well-trained staff inquiring via a face-to-face interview and recording the information, including age, gender, living area, income level, educational attainment, current drinking status, current smoking status, physical activity, etc.

### 2.3. Dietary Assessment

Dietary information was assessed by a widely used semi-quantitative FFQ to assess habitual food intake from the preceding year [23]. This FFQ, composed of 64 food items, was designed to investigate the dietary intake of Chinese residents. The 64 food items varied from rice and its products, wheat and its products, soybean and its products, vegetables, fruits, dairy and its products, meats, poultry, aquatic products (including sea food, freshwater fish, shellfish and mollusks), eggs, fried dough foods, snacks and nuts, and other kinds of food. To assess fat intake and energy intake, the consumption of cooking oil in the last 30 days was also investigated. The semi-quantitative FFQ questionnaire used in this survey includes three questions for each food: (1) have you consumed this food or not; (2) the frequency of consumption of this food (daily, weekly, monthly, or yearly); (3) the weight of edible intake of each food (gram each time). The weight of food intake was recorded and converted into edible intake by an experienced investigator. The daily eating weight was calculated based on frequency. We summed up each person’s daily energy and macronutrient intake from their daily food, based on the China Food Composition Table (2009) and the China Food Composition (2018) document [24,25].

### 2.4. Anthropometric Measurements

Anthropometrical measurements were conducted by well-trained health workers at the local community health center who followed a reference protocol recommended by the National Health Commission of the People’s Republic of China [26]. Anthropometric measurements included barefoot standing height, weight with minimal clothing (on a digital, electronic scale), and waist circumference (in the horizontal plane at a point marked just above the right ilium on the mid axillary line at minimal respiration). Height and waist circumference were recorded to the nearest 0.1 cm. Body mass index (BMI) was calculated by weight (Kg)/height^2^ (m^2^). An electronic weight scale (TANITA HD-390, Shanghai, China) and a TZG height meter were used in this surveillance.

### 2.5. Laboratory Test

The fasting blood sample was collected by an evacuated blood collection tube during the physical examination. The serum was separated via centrifugation (1500× *g*, 15 min). The concentrations of triglycerides (TG; mmol/L), total cholesterol (TC; mmol/L), low density lipoprotein cholesterol (LDL-c; mmol/L), high density lipoprotein cholesterol (HDL-c; mmol/L), and fasting plasma glucose (FPG; mmol/L) in the serum were analyzed by the GPO-HMMPS glycerol blanking method, cholesterol oxidase-HMMPS method, direct determination-selective protection method and direct determination-antibody block method, and the hexokinase G-6-PDH method, respectively. The concentrations of hemoglobin A1C (HbA_1c_; %) in whole blood collected by the evacuated blood collection tube containing EDTA-K2 were determined by high-performance liquid chromatography, and the testing method was certified by the International Federation of Clinical Chemistry (IFCC). All the laboratories that participated in this surveillance have passed the external quality assessment of the National Center for Clinical Laboratories.

### 2.6. Definition of Glycemic Control and Impaired Fasting Glucose

Hemoglobin A1C (HbA_1c_) is considered as the gold standard for the monitoring and treatment of diabetes, it indicates an average of blood glucose levels over the past 3 months. In the present study, according to the *Guideline for the Prevention and Treatment of Type 2 Diabetes Mellitus in China (2020 edition)* [27], glycemic control in diabetic patients was defined as HbA_1c_ less than 7%. Impaired fasting glucose was defined as fasting plasma glucose greater than 7 mmol/L.

### 2.7. Covariates

The variety of variables that were used for multiple adjustments in the logistic regression analysis were defined as follows. (1) Age (in years) was categorized into four categories: 18–44, 45–59, 60–74, 75 and above. (2) Living areas were separated into urban and rural. (3) Income level (RMB/year) was categorized into four categories according to its distribution into quartiles: <6666.66, 6666.66–15,000, 15,000–25,000, and >25,000. (4) Educational attainment was divided into primary school or below, middle school and high school, and college or higher. (5) Body mass index (BMI) was categorized as underweight (BMI < 18.5 Kg/m^2^), normal (18.5 Kg/m^2^ ≤ BMI < 24 Kg/m^2^), overweight (24 Kg/m^2^ ≤ BMI < 28 Kg/m^2^), and obese (BMI ≥ 28 Kg/m^2^) [28]. (6) Occupational physical activity was categorized into three levels: light (defined as sitting 75% of the time or standing 25% of the time for special occupational activities, such as office workers), middle (defined as sitting 25% of the time or standing 75% of the time for special occupational activities, such as motor vehicle drivers), and heavy (defined as 40% sitting or 60% standing for special occupational activities such as dock workers). (7) Physical activity was calculated based on the total weekly metabolic equivalent (MET) and the total weekly duration of exercise at different levels: low (MET < 600), moderate (600 ≤ MET ≤ 3000), or high (MET > 3000) [29]. (8) Total energy intake was calculated from the FFQ and separated into quartiles in each gender (labelled as Q1, Q2, Q3, and Q4). (9) Current smoking status was categorized as yes or no, regardless of smoking history. (10) Current drinking status was categorized as yes or no in the last 30 days. Diabetes medication use, insulin use, following diabetic diets, increased exercise, and glucose monitoring were assessed by self-reported questionnaires.

### 2.8. Dietary Pattern

RRR was used to derive dietary patterns (DP) and was first introduced by Hoffmann et al. [21], and this method has been widely used to explore the association between DPs and chronic disease [30,31]. In the RRR procedure, food groups were set as predictor variables. Before the RRR procedure, 64 single food items were combined into 24 food groups based on their similarities in nutrient profiles and habitual practices. When calculating the intake of milk and dairy products (including fresh milk, milk powder, cheese, etc.), the weight of each dairy product was converted into the weight of fresh milk, using the protein content in fresh milk as the standard, and the summary of the converted weight of each dairy products was the intake of milk and dairy products. The intake of soybean and its products (including soy milk, tofu, and yuba) was calculated in the same way, and the conversion standard is the protein content in dry soybeans. The intakes of other food groups were calculated as a summary of the weight of the raw food (e.g., fruit, vegetables) or the edible weight of the food (e.g., fish, shrimp, crab, etc.). Glycemic control was assessed by HbA_1c_ concentration, which was chosen as the dependent variable to derive DPs. The HbA_1c_ concentration was Box-Cox transformed before the RRR procedure to improve normality. In order to improve the clinical relevance and interpretation of these DPs, food groups with absolute factor loadings of 0.25 were selected to construct simplified DP scores. The factor loading of each food group represents the size and direction of the contribution to the DP; a positive or negative factor loading represents a high or low frequency of consumption. The sum of the factor loading of each food group multiplied by the normalized intake frequency (gram per day) of this food group is the DP score of each participant. For every participant, the higher the DP score, the closer the diet was to that DP. The DP scores were then divided into quartiles for further analyses.

Considering the differences in consumption levels of certain food groups between males and females, we split our data by gender and conducted RRR and further analyses on males and females, respectively. 

### 2.9. Assessment of Macronutrients and Dietary Fiber Intake

Macronutrients (carbohydrates, protein, and fat) and dietary fiber intake were assessed by FFQ. In calculating the fat intake, condiment consumption was used. Macronutrients and dietary fiber intake were evaluated, the cut-off value was defined as according to the *Guideline for the Prevention and Treatment of Type 2 Diabetes Mellitus in China (2020 edition)* [27] for the above nutrients. Then, we calculated and compared the proportion of participants with each dietary pattern who met the recommendations for macronutrients and dietary fiber. For dietary fiber intake, the recommendation was 14 g/ 1000 kcal. The recommendation for the percentage of macronutrient intake from energy intake according to the *Guideline for the Prevention and Treatment of Type 2 Diabetes Mellitus in China (2020 edition)* [27] is shown in Appendix A.

### 2.10. Statistical Analysis

The continuous variables with abnormal distribution were presented by median and interquartile range (M (*P_25_, P_75_*)), counts, and percentages for the categorical variables. The Kruskal–Wallis test was used for the comparison of two groups with abnormal distribution data. The Chi-square test was used for the comparison of prevalence (including impaired fasting glucose and glycemic control) between the two groups. The associations between the DP scores and the risk of poor glycemic control were estimated using multiple-variable-adjusted logistic regression models, and the results were shown as odds ratios (*OR*s). The models were adjusted for age, living area, income levels, educational attainment, body mass index, occupational physical activity level, physical activity status, total energy intake level, current smoking status, current drinking status, diabetes medication use, insulin use, following diabetic diets, increased exercise, and glucose monitoring. All analyses were conducted separately for males and females. Statistical data cleaning and analysis were performed on SAS software (version 9.4, SAS Institute Inc, Cary, NC, USA), plot drawing was performed on RStudio (version 4.1.2). DP analysis was performed on SAS (version 9.4; SAS Institute, Cary, NC, USA) by the SAS PLS procedure with the method = RRR option. *p* < 0.05 was considered statistically significant.

## 3. Results

### 3.1. Characteristics of Participants

The characteristics of 2031 participants are displayed in Table 1. The number of females was higher than that of males (55.2% vs. 44.8%). The comparison of income levels, living areas (urban or rural), education attainment, physical activity, current smoking status, and current drinking status between males and females showed significant differences (*p* < 0.05). As for clinical indicators, the median level in female participants tended to be less than in male participants except for fasting plasma glucose and HbA_1c_. As for anthropometric indicators, male participants had a higher waist circumference (WC) and a smaller waist to height ratio (WHtR) than females (*p* < 0.05).

### 3.2. Prevalence of Glycemic Control

Table 2 shows the prevalence of glycemic control in the different categories; the statistical inference was completed by the Chi-square test. The prevalence of glycemic control among all the participants was 61.5% in general. Generally, the prevalence of glycemic control was significantly different for the different categories of age group, income level, living area, and education attainment, but was not significantly different for the different categories of physical activity level, current smoking status and current drinking status. No statistical significance was observed when comparing the prevalence among male and female participants in all categories. Overall, older adults aged 75 years and older had the highest glycemic control rate at 72.6%, and the underweight group, measured by BMI, had the lowest glycemic control rate at 42.9%, for males the rate was 30%.

### 3.3. Dietary Patterns among Chinese Diabetics in This Study

Figure 1 shows the factor loading of DPs based on 24 kinds of food item for male and female participants, respectively. Two gender-specific simplified DP scores were derived using RRR which were positively associated with HbA_1c_ concentration. For both males and females, these DPs were unhealthy, and were characterized by a low intake of freshwater fish, poultry, and fruits. Additionally, the male DP was characterized by a high intake of wheat and its products, and a low intake of vegetables, corn and its products, shrimp and crab, and beans, while the female DP was characterized by a low intake of snacks and nuts, and algae and mushroom.

### 3.4. Food, Macronutrients, and Dietary Fiber Intake among Chinese Diabetics

Table 3 shows the intake of 24 food groups by participants in different DP-score groups in this study. For male participants in this study, the intake of meat, processed meat, internal organs, mollusks, eggs (except pidan), beverages, juice, and coffee was not significantly different between the quartile groups of DP scores. For female participants, the intake of beverages was not significantly different between the quartile groups of DP scores. For half of the participants in this study, the intake of beans, fried dough foods, processed meat, internal organs, marine fish, shrimp and crab, mollusks, and beverages was less than 1 g/day.

Table 4 shows the macronutrient intake (g/day) and percentage of energy from macronutrients (%) of the participants grouped according to the interquartile range of DP scores. For male participants, the median intake of energy, carbohydrate, protein, and fat was 2587.2 kcal/day, 382.8 g/day, 71.2 g/day, and 74.5 g/day, respectively. The median percentage of energy from carbohydrate intake, protein intake, and fat intake was 63.5%, 11.2%, and 26.2%, respectively. The intake of energy and carbohydrates per day by male participants in the Q4 group was the highest among these four groups, and the intake of protein by male participants in the Q1 group was the highest among these four groups. The percentage of energy from protein intake for male participants in the Q1 group was the highest among these four groups. For female participants, the median intake of energy, carbohydrate, protein, and fat was 2254.3 kcal/day, 359.1 g/day, 62.0 g/day, and 64.0 g/day, respectively. The median percentages of energy from carbohydrate intake, protein intake, and fat intake were 64.1%, 10.9%, and 27.1%, respectively. The intake of energy, carbohydrate, and protein per day by female participants in the Q4 group was the highest among these four groups, and the intake of fat by female participants in the Q1 group was the highest among these four groups. The percentage of energy from protein intake and the percentage of energy from fat intake for female participants in the Q1 group were the highest among these four groups.

Table 5 shows the proportion of participants from the different dietary pattern-score groups who reached the recommended intake level of macronutrients according to the *Guideline for the Prevention and Treatment of Type 2 Diabetes Mellitus in China (2020 edition)* [27].

Table 6 shows dietary fiber intake among Chinese diabetics according to quartiles of dietary pattern scores. For males and females, fiber intake per 1000 kcal was 6.0 g and 6.2 g, respectively. For all participants, fiber intake in the first quartile groups was highest among these four groups.

### 3.5. Dietary Pattern Score, an Anthropometric and Clinical Indicator

DP scores were calculated for male and female participants separately. Male and female participants were divided into four groups according to the interquartile range of DP scores. Table 7 shows the clinical and anthropometric indicators of the participants in this study by gender and in different groups. For the male participants, there were significant differences in fasting plasma glucose (FPG), HbA_1c_, TC, and LDL levels among the four groups (*p* < 0.05). The medians of FPG, HbA_1c_, TC, and LDL levels of male participants in the Q4 group were 8.06 mmol/L, 7.2%, 4.75 mmol/L, and 3.02 mmol/L, respectively, which were higher than those of the other three groups. Compared with the participants in the Q1 group, those in the Q4 group had a 9.2% increase in FPG, an 18.0% increase in HbA_1c_, a 5.6% increase in TC, and an 8.6% increase in LDL. Among these four groups, there were significant differences in the prevalence of glycemic control and the prevalence of impaired fasting glucose (*p* < 0.05). The prevalence of glycemic control in the Q4 group was 45.6%, the lowest among the four groups; the prevalence of impaired fasting glucose in the Q4 group was 72.1%, the highest among the four groups.

For female participants, there were significant differences in fasting plasma glucose (FPG) and HbA_1c_ levels among the four groups (*p* < 0.05). The medians of FPG and HbA_1c_ levels in female participants in the Q4 group were 7.90 mmol/L and 6.8%, which were the highest among these four groups. Compared with the participants in the Q1 group, those in the Q4 group had an 8.2% increase in FPG and a 9.7% increase in HbA_1c_. Among these four groups, there were significant differences between the prevalence of glycemic control and impaired fasting glucose (*p* < 0.05). The prevalence of glycemic control in the Q4 group was 52.4%, the lowest among these four groups; the prevalence of impaired fasting glucose in the Q4 group was 68.2%, the highest among the four groups.

### 3.6. Association between Dietary Patterns and Risk of Poor Glycemic Control

Associations between DPs and the risk of poor glycemic control were assessed by logistic regression and calculated by gender, respectively. For male and female participants, four groups were divided according to the interquartile range of their DP scores, with the Q1 group as the reference.

For male participants, the Q4 group had a positive association with the risk of poor glycemic control (Q4 vs. Q1, *OR* = 2.69, 95% *CI*: 1.76 to 4.10, *p* < 0.0001) in the multiple-variable-adjusted logistic regression model; the results are shown in Figure 2a. Meanwhile, a negative association was observed when describing the relationship between diabetic medication use, insulin use, and glycemic control. When assessing the association between increased exercise and glycemic control, the risk of poor glycemic control for participants not increasing their level of exercise was nearly 70% higher than the participants increasing their level of exercise (“No” vs. “Yes”, *OR* = 1.68, 95% *CI*: 1.22 to 2.31, *p* < 0.0001). 

For female participants, the Q3 and Q4 groups had a positive association with the risk of poor glycemic control (Q3 vs. Q1, *OR* = 1.82, 95% *CI*: 1.25 to 2.65, *p* < 0.0001; Q4 vs. Q1, *OR* = 2.18, 95% *CI*: 1.48 to 3.20, *p* < 0.0001) in the multiple-variable-adjusted logistic regression model; the results are shown in Figure 2b. A negative association was observed when describing the relationship between insulin use and glycemic control. When assessing the association between living area and glycemic control, the risk of poor glycemic control for the participants living in rural areas was 52% higher than for the participants living in urban areas (“rural” vs. “urban”, *OR* = 1.52, 95% *CI*: 1.13 to 2.05, *p* < 0.0001).

## 4. Discussion

Strong evidence supports the effectiveness of medical nutrition therapy (MNT) interventions provided by registered dietitian nutritionists for improving HbA_1c_ levels, with absolute decreases up to 2.0% (in type 2 diabetes) and up to 1.9% (in type 1 diabetes) at 3–6 months. HbA_1c_ reduction from MNT is equal to or higher than the expected value of using currently available medications for type 2 diabetes [32,33]. For MNT, DP was an effective and widely used approach. Among many DPs that can improve glycemic control in diabetes, the Mediterranean diet is well-documented and effective. However, the Mediterranean diet cannot not be used everywhere; for every country, the DP for MNT should be composed of local food. Few studies have explored the association between DPs and glycemic control in Chinese diabetes [34]. The current research examined two poor and gender-specific simplified DPs among 2031 Chinese diabetic samples aged 18 and above, which were characterized by a low intake of freshwater fish, poultry, and fruits for all diabetes. For male participants, the gender-specific DP was characterized by a high intake of wheat and its products, a low intake of vegetables, corn and its products, as well as shrimp and crab, and beans. For female participants, the gender-specific DP was characterized by a low intake of snacks and nuts, as well as algae and mushroom. In this unhealthy DP, higher DP scores indicated better adherence to the DP. High DP scores were associated with an increased risk of poor glycemic control after adjustment; *OR*s for the fourth quartile (Q4) using the first quartile as a reference, were 2.69 (95% *CI*: 1.76 to 4.10) for males and 2.18 (95% *CI*: 1.48 to 3.20) for females, respectively. When calculating the clinical and anthropometric indicators of participants, it was found that, for males, HbA_1c_ was 1.1% higher in Q4 than in Q1, while for females, HbA_1c_ was 0.6% higher in tQ4 than in Q1.

Similar results among Chinese participants were obtained in Huang’s study, which found that an increased intake of vegetables and fish improved the metabolic status of diabetics. Compared with the high-fat meat diet and traditional Chinese food-snack diet, diabetics who follow a vegetable and fish diet have lower levels of HbA_1c_ and fasting plasma glucose [35]. In another cross-sectional study conducted among Chinese non-diabetic participants aged 45–59, compared with those in the first quartile, participants in the fourth quartile of the grains–vegetables pattern score had a lower *OR* (0.82, 95% *CI*: 0.72 to 0.94) for glycemic control. In this study, it was found that for Chinese diabetic patients, less fruit consumption was detrimental to their glycemic control. That is consistent with the results of Su’s study. In Su’s study, 19,473 Chinese diabetic patients were investigated. Compared with patients who did not consume fresh fruits, the risk of poor glycemic control in patients with fresh fruit consumption frequencies of 1–4 times/week and ≥5 times/week decreased by 20% and 30%, respectively. Patients with fruit consumption of 50–99 g/day and ≥100 g/day had a lower risk of poor glycemic control, with *OR*s (95% *CI*) of 0.71 (95% *CI*: 0.62 to 0.83) and 0.68 (95% *CI*: 0.59 to 0.78), respectively [36]. These observable effects of plant-based diets may be related to their intrinsic properties, including their high-fiber content and the presence of antioxidant and anti-inflammatory micronutrients (such as polyunsaturated fatty acids, polyphenols, and magnesium, etc.), and they have a positive effect on the prevention of microvascular and macrovascular complications [37,38]. 

The effect of snacks on glycemic control in diabetic patients may be related to the timing of snacking and the type of snacking. In one crossover randomized clinical trial conducted in Japan, the participants consumed an identical breakfast, lunch, dinner, and snacks throughout the test period. They consumed breakfast, lunch, and dinner at the same time, but the snack time was different. It was found that consuming snacks in the mid-afternoon (3.5 h after lunch) led to significantly lower mean amplitudes of glycemic excursions and incremental areas under the curve for glucose after dinner, compared with snacking just after lunch (0.5 h after lunch), which can be explained by the difference in the total amount of carbohydrate intake and the Staub-Traugott (second-meal) effect [39]. In a randomized controlled trial using a crossover design, diabetic patients who consumed a low-carbohydrate bedtime snack (egg) had a lower fasting glucose level (7.6 ± 0.2 mmol/L vs. 8.2 ± 0.3 mmol/L; *p* = 0.02) and a lower fasting plasma insulin level (111 ± 52 pmol/L vs. 128 ± 56 pmol/L; *p* = 0.04) when compared to those who consumed a high-carbohydrate protein-matched and isocaloric bedtime snack (yogurt) [40]. In one systematic literature review including four studies that used corn starch, it was reported that the low-dose use of corn starch seemed to have benefits over a high-dose use of corn starch in terms of improved nocturnal and fasting glucose concentrations [41]. Since this study did not investigate the macronutrient composition of snacks consumed by the female participants and the timing of their consumption in the survey, it cannot answer why snack consumption had a positive effect on glycemic control in the female participants in this study. In subsequent cross-sectional studies exploring the effect of snacking on glycemic control in diabetic patients, more detailed investigations on the timing of snacking and the number of macronutrients in snacks are needed.

When exploring DPs among the diabetics in this study, we selected food groups with absolute factor loadings of 0.25 to construct a simplified DP and describe the characteristics of that. Therefore, a high intake of fried dough food was not included in the food groups of the DP in this study (for male participants, the factor loadings for fried dough food were 0.2043 and for female participants, the factor loadings for fried dough food were 0.1868). Fried dough food is still a risk factor for poor glycemic control in both males and females with diabetes. Deep-fried dough sticks and oil cakes are the traditional Chinese breakfast. However, they are unhealthy, as they are made from refined wheat flour and cooked through deep frying, which results in excessive fat intake. Excessive fat intake increases the amount of free fatty acid in the blood, increases the risk of obesity, and causes insulin resistance; the mechanism might include the generation of lipid metabolites (diacylglycerol), proinflammatory cytokines (TNF-α, IL1β, IL6, MCP1), and cellular stress, including oxidative and endoplasmic reticulum stress [42]. In addition, a survey showed that people are facing a health risk of exposure to aluminum (Al) from this traditional food through its food additives (alum, etc.). For people in northern China, deep-fried dough sticks are the main Al contributor, providing 28.2% of the daily intake [43]. An experimental study which aimed to investigate the effects of Al exposure on glucose metabolism and its mechanism showed that Al exposure was found to affect glucose metabolism through a combination of pancreatic damage and the reduction of glucose transporter protein 4 expression in the skeletal muscle [44]. These may explain the relationship between fried pasta intake and the risk of poor glycemic control in diabetic patients.

We partially explained the possible reasons for the effect of DP on glycemic control in diabetic patients in the present study from the perspective of food groups in the previous section. To better explain this finding, we analyzed the daily macronutrient intake and the percentage of energy from macronutrient intake of participants in different DP-score groups. We found that dietary protein intake and the percentage of energy from protein intake (PEPI) were lower for diabetic patients in the highest quartile of DP score compared to those in the lowest quartile of DP score. In addition, in the highest quartile of DP score, the ratio of diabetic patients whose PEPI could not reach the recommended level of the *Guideline for the Prevention and Treatment of Type 2 Diabetes Mellitus in China (2020 edition)* [27] was 95.6%. Lower protein intake may lead to the increased loss of lean body mass in diabetic patients [45], which can be even more detrimental to the glycemic control of diabetics in the long run [46]. A meta-analysis that included 13 studies found that high protein diets (protein to energy ratio between 25% and 33%) did not significantly reduce HbA_1 c_ levels (%) in patients with diabetes over a period of 12 weeks to 52 months (−0.05 (95% *CI* (−0.18 to 0.08, *p* = 0.92))) [17]. Therefore, we also share the view that the level of protein intake and the percentage of energy from protein intake for diabetic patients should be recommended in an appropriate range [47]. Although a meta-analysis showed that plant protein is better for glycemic control for diabetic patients than animal protein [48], this may be related to the richness of phytochemicals and dietary fiber in the plant [49]. In the present study, with HbA_1c_ as the response variable, a lower intake of poultry and freshwater fish was detrimental to glycemic control in patients with diabetes. This has some similarities with the Mediterranean diet recommendations for the intake of animal proteins, which recommend replacing processed meat and red meat with fish and poultry.

For diabetic patients, a good dietary pattern plays an equally positive role in liver fibrosis, cognitive ability, lipid metabolism, and systolic and diastolic blood pressure. According to one study of 160 diabetic patients with non-alcoholic fatty liver disease (NAFLD) confirmed by biopsy which used the Prevencion con Dieta Mediterranea (PREDIMED) questionnaire of 14 items used to assess the participants’ adherence to the Mediterranean diet, adherence to the Mediterranean diet was a protective factor against liver fibrosis (*OR* = 0.6, 95% *CI*: 0.4 to 0.8; *p* = 0.01). This may be attributable to the improvement of the cardiometabolic profile due to the Mediterranean diet and the content of antioxidant compounds such as polyphenols and vitamin E from olive oil and fruits in the Mediterranean diet [50]. In a cross-sectional study conducted in Germany targeting diabetics, it was found that closer adherence to the Mediterranean diet was associated with better verbal memory performance in patients with type 2 diabetes with a known diabetes duration ≥5 years; this may be related to the richness of *n*-3 fatty acids in the Mediterranean diet [51]. The results of a meta-analysis that included nine studies with 1178 patients showed that, compared with control diets, Mediterranean diets led to decreased concentrations of total cholesterol and triglyceride (−0.14 mmol/l; *CI*, −0.19 to −0.09 and −0.29 mmol/l; *CI*, −0.47 to −0.10, respectively) and increased concentrations of high-density lipoprotein (0.06 mmol/l; *CI*, 0.02 to 0.10). A Mediterranean DP was also associated with a decline of 1.45 mm Hg (*CI*, −1.97 to −0.94) for systolic blood pressure and 1.41 mm Hg (*CI*, −1.84 to −0.97) for diastolic blood pressure [13].

The highlight of this study is that the other risk factors affecting glycemic control in diabetic patients were considered as comprehensively as possible, and the relationship between DP and the risk of poor glycemic control in diabetic patients was explored by a logistic regression model with multiple-variable correction based on the available data. In the process, we also assessed the other factors affecting glycemic control in diabetic patients, regarding which we would like to discuss three points here. First, the participants living in rural areas had a higher risk (*OR* = 1.21, 95% *CI*: 0.86 to 1.71 for males and *OR* = 1.52, 95% *CI*: 1.13 to 2.05 for females) of poor glycemic control compared with those living in urban areas. This result is consistent with the findings of Wang’s previous study, and this difference may be related to environmental, socioeconomic, lifestyle, and policy factors [2]. Second, participants were interviewed about their measures of glycemic control, including the question, “Do you control your blood glucose by taking more exercise?” In the present study, a higher risk of poor glycemic control (*OR* = 1.68, 95% *CI*: 1.22 to 2.31) was found for male participants who did not take more exercise, but for female participants, whether or not they took more exercise was not associated with the risk of poor glycemic control. This may be explained by a study that explored the association between HbA_1c_ and exercise in diabetic patients. In one randomized controlled trial conducted in the USA, it was found that HbA_1c_ levels were reduced in the group that undertook combined resistance and aerobic training after a 9-month exercise program, with a mean change of −0.34% (95% *CI*: −0.64% to 0.03%; *p* = 0.03). However, the mean changes in HbA_1c_ were not statistically significant in either the resistance training (−0.16%; 95% *CI*: −0.46% to 0.15%; *p* = 0.32) or the aerobic (−0.24%; 95% *CI*: −0.55% to 0.07%; *p* = 0.14) groups compared with the control group. This may be attributed to the effect of resistance training in preventing lean mass loss and producing weight loss in combination with aerobic exercise [52]. To lower glycated hemoglobin, a combination of aerobic and anaerobic exercise should be recommended for people with diabetes. The American College of Sports Medicine makes exercise training recommendations for people with diabetes that include the type, intensity, frequency, and duration of exercise. Compared with the recommendations for non-diabetic people, the type of exercise training was classified into four categories: aerobic, resistance, flexibility, and balance, and the recommended frequency of the above exercise was 3–7 days per week to 2–3 days per week or more. The recommended intensity for aerobic exercise was 40% to 59% of oxygen uptake reserve (VO_2_R) or heart rate reserve (HRR) (moderate), with a rating of perceived exertion (RPE) of 11 to 12; or 60% to 89% of VO_2_R or HRR (vigorous), with a rating of perceived exertion of 14 to 17. The recommended intensity for resistance exercise was moderate at 50% to 69% of one repetition maximum (1RM), or vigorous at 70% to 85% of 1RM. For flexibility exercises, the recommended intensity was to stretch to the point of tightness or slight discomfort. It is also claimed that increased postprandial energy expenditure lowers the glucose levels regardless of exercise intensity or type, and that 45-min durations provide the most consistent benefits [53]. Third, this study found that following a diabetic diet was not associated with glycemic control in both male and female participants, which may be related to a lack of nutrition knowledge, not being prescribed a diabetic diet by physicians, and not following a diabetic diet in daily life [54,55].

In this study, the dietary status of Chinese diabetics aged 18 years and older, including food consumption, macronutrient intake, and dietary fiber intake, was investigated using a standardized FFQ questionnaire. The association between DP and glycemic control in diabetic patients was clarified by a multiple-variable-adjusted logistic regression model. However, there were still some limitations that should be elaborated. First, due to the limitations of the cross-sectional study, it was difficult to show the causal relationship between DPs and glycemic control. Second, medication use was specified as binary variables, merely asking diabetic patients whether they have taken diabetic medications does not capture the degree of medication adherence, and the association between DPs and glycemic control may be influenced by medication adherence. Third, in the questionnaire, we did not collect information on the type of exercise taken by diabetic patients and could not classify physical exercise into the categories, aerobic, resistance, stretching, and balance, so it may be hard to assess the specific role of exercise in glycemic control in Chinese diabetics. In a further study, we will investigate the types and amounts of edible oils in more detail and improve the database of fatty acids more comprehensively for Chinese edible oils and foods, to provide more evidence for the mechanistic exploration of dietary patterns. Besides considering the effect of lean body mass on glycemic control in diabetic patients, we will also consider measuring body composition in diabetic patients in subsequent studies as funding allows. Nevertheless, to the best of our knowledge, the present study is one of the few studies in recent years that has explored the association between DPs and glycemic control in community-dwelling diabetic patients throughout China.

## 5. Conclusions

Among the Chinese diabetic patients in the current study, using RRR, two gender-specific simplified DP scores were derived which were positively associated with HbA_1c_ levels. For both males and females, these DPs were unhealthy, and were characterized by the low intake of freshwater fish, poultry, and fruits. Additionally, the male DP was characterized by the high intake of wheat and its products, and the low intake of vegetables, corn and its products, shrimp and crab, and beans, while the female DP was characterized by the low intake of snacks and nuts, and algae and mushroom. The results indicate that targeted dietary advice for diabetic patients may improve their glycemic control.

## Figures and Tables

**Figure 1 nutrients-15-00056-f001:**
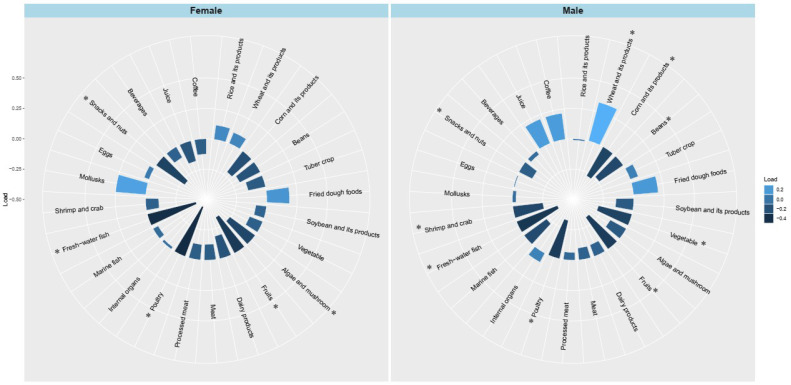
Factor loading of food groups for male and female participants. * Represents an absolute value of factor loading of this food group larger than 0.25.

**Figure 2 nutrients-15-00056-f002:**
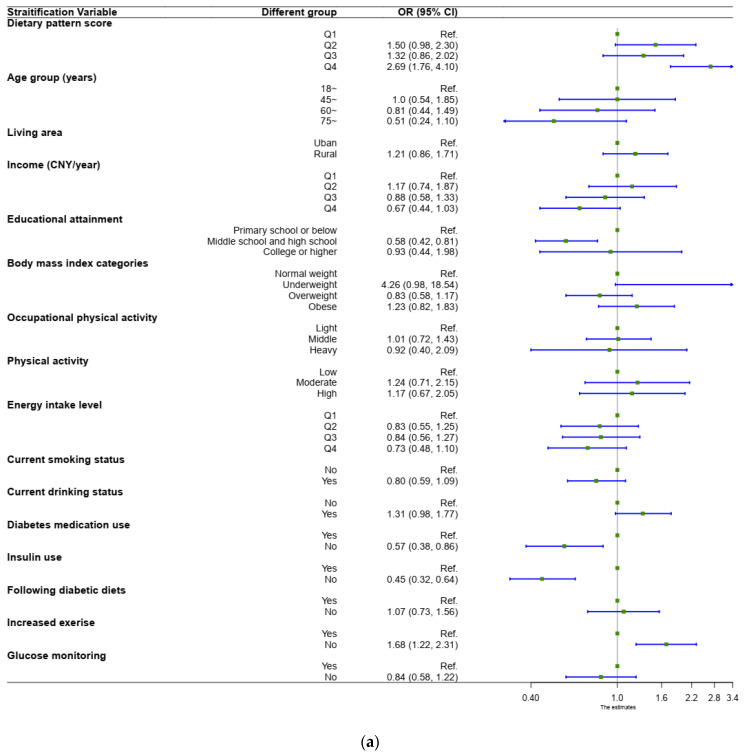
(**a**) Multiple-variable-adjusted association between dietary pattern and the risk of poor glycemic control for male participants. (**b**) Multiple-variable-adjusted association between dietary pattern and the risk of poor glycemic control for female participants.

**Table 1 nutrients-15-00056-t001:** Demographic and clinical characteristics of study participants (*n*= 2031).

		Male	Female	Total	Statistics and *p*-Value
*n* (%)		910 (44.8)	1121 (55.2)	2031 (100.0)	*χ*^2^ = 21.92, *p* < 0.0001
Age (years)	18~	60 (6.6)	38 (3.4)	98 (4.8)	*χ*^2^ = 17.15, *p* = 0.0007
	45~	291 (32.0)	358 (31.9)	649 (32.0)	
	60~	458 (50.3)	629 (56.1)	1087 (53.5)	
	75~	101 (11.1)	96 (8.6)	197 (9.7)	
Income level (*n*, %)	Q1	209 (23.0)	339 (30.2)	548 (27.0)	*χ*^2^ = 24.00, *p* < 0.0001
	Q2	137 (15.1)	207 (18.5)	344 (16.9)	
	Q3	236 (25.9)	229 (20.4)	465 (22.9)	
	Q4	328 (36.0)	346 (30.9)	674 (33.2)	
Living area (*n*, %)	Urban	624 (68.6)	672 (60.0)	1296 (63.8)	*χ*^2^ = 16.18, *p* < 0.0001
	Rural	286 (31.4)	449 (40.1)	735 (36.2)	
Position (*n*, %)	East	469 (51.5)	535 (47.7)	1004 (49.4)	*χ*^2^ = 4.66, *p* = 0.0975
	Middle	246 (27.0)	351 (31.3)	597 (29.4)	
	West	195 (21.4)	235 (21.0)	430 (21.2)	
Education attainment (*n*, %)	Primary school or below	294 (32.3)	687 (61.3)	981 (48.3)	*χ*^2^ = 174.00, *p* < 0.0001
	Middle school	573 (63.0)	418 (37.3)	991 (48.8)	
	College or higher	43 (4.7)	16 (1.4)	59 (2.9)	
Physical activity (*n*, %)	Low	73 (8.0)	85 (7.6)	158 (7.8)	*χ*^2^ = 59.46, *p* < 0.0001
	Moderate	408 (44.8)	324 (28.9)	732 (36.0)	
	High	429 (47.1)	712 (63.5)	1141 (56.2)	
Current smoking status (*n*, %)	No	542 (59.6)	1078 (96.2)	1620 (79.8)	*χ*^2^ = 416.92, *p* < 0.0001
	Yes	368 (40.4)	43 (3.8)	411 (20.2)	
Current drinking status (*n*, %)	No	445 (48.9)	984 (87.8)	1429 (70.4)	*χ*^2^ = 364.02, *p* < 0.0001
	Yes	465 (51.1)	137 (12.2)	602 (29.6)	
Body mass index categories (*n*, %)	Normal weight	243 (26.7)	359 (32.0)	602 (29.6)	*χ*^2^ = 11.37, *p* = 0.0099
	Underweight	10 (1.1)	18 (1.6)	28 (1.4)	
	Overweight	437 (48.0)	461 (41.1)	898 (44.2)	
	Obese	220 (24.2)	283 (25.3)	503 (24.8)	
Glycemic control (*n*, %)		558 (61.3)	690 (61.6)	1248 (61.5)	*χ*^2^ = 0.01, *p* = 0.9144
Physical activity, MET-min/w		2784.8 (1362.0, 6312.0)	4404.0 (1945.5, 8481.0)	3666.0 (1602.0, 7547.0)	*χ*^2^ = 43.49, *p* < 0.0001
Age, years		63.1 (55.0, 70.0)	63.1 (57.2, 69.0)	63.1 (56.2, 69.5)	*χ*^2^ = 0.10, *p* = 0.7517
WC, cm		90.4 (84.2, 96.9)	86.1 (80.1, 93.0)	88.2 (82.0, 95.0)	*χ*^2^ = 76.97, *p* < 0.0001
WHtR		0.54 (0.51, 0.58)	0.56 (0.52, 0.60)	0.55 (0.51, 0.59)	*χ*^2^ = 38.75, *p* < 0.0001
BMI, kg/m^2^		25.78 (23.69, 27.96)	25.45 (23.15, 28.04)	25.58 (23.34, 27.98)	*χ*^2^ = 2.13, *p* = 0.1441
FPG, mmol/L		7.78 (6.34, 9.93)	7.54 (6.10, 9.67)	7.67 (6.20, 9.83)	*χ*^2^ = 2.66, *p* = 0.1027
HbA_1c_, %		6.5 (5.5, 7.8)	6.4 (5.5, 7.8)	6.4 (5.5, 7.8)	*χ*^2^ = 0.00, *p* = 0.9663
TC, mmol/L		4.61 (3.93, 5.30)	5.05 (4.35, 5.77)	4.82 (4.15, 5.55)	*χ*^2^ = 76.68, *p* < 0.0001
TG, mmol/L		1.49 (1.01, 2.30)	1.53 (1.07, 2.32)	1.51 (1.04, 2.32)	*χ*^2^ = 1.93, *p* = 0.1648
LDL-C, mmol/L		2.91 (2.37, 3.54)	3.19 (2.59, 3.87)	3.06 (2.49, 3.73)	*χ*^2^ = 43.93, *p* < 0.0001
HDL-C, mmol/L		1.07 (0.90, 1.25)	1.18 (1.01, 1.40)	1.13 (0.95, 1.34)	*χ*^2^ = 83.05, *p* < 0.0001

**Table 2 nutrients-15-00056-t002:** Prevalence of glycemic control among Chinese diabetics aged 18 years and above in this study (*n*, %).

		Male	Female	Total	Statistics and *p*-Value
*n* (%)		558 (61.3)	690 (61.6)	1248 (61.5)	*χ*^2^ = 0.01, *p* = 0.9144
Age (years)	18~	34 (56.7)	24 (63.2)	58 (59.2)	*χ*^2^ = 0.41, *p* = 0.5241
	45~	170 (58.4)	219 (61.2)	389 (59.9)	*χ*^2^ = 0.51, *p* = 0.4764
	60~	282 (61.6)	376 (59.8)	658 (60.5)	*χ*^2^ = 0.36, *p* = 0.5500
	75~	72 (71.3)	71 (74.0)	143 (72.6)	*χ*^2^ = 0.18, *p* = 0.6744
	Statistics and *p*-value	*χ*^2^ = 5.82, *p* = 0.1206	*χ*^2^ = 7.14, *p* = 0.0675	*χ*^2^ = 11.54, *p* = 0.0091	
Income level	Q1	112 (53.6)	198 (58.4)	310 (56.6)	*χ*^2^ = 1.22, *p* = 0.2690
	Q2	70 (51.1)	113 (54.6)	183 (53.2)	*χ*^2^ = 0.40, *p* = 0.5249
	Q3	145 (61.4)	138 (60.3)	283 (60.9)	*χ*^2^ = 0.07, *p* = 0.7946
	Q4	231 (70.4)	241 (69.7)	472 (70.0)	*χ*^2^ = 0.05, *p* = 0.8265
	Statistics and *p*-value	*χ*^2^ = 22.78, *p* < 0.0001	*χ*^2^ = 15.41, *p* = 0.0015	*χ*^2^ = 36.41, *p* < 0.0001	
Living area	Urban	407 (65.2)	446 (66.4)	853 (65.8)	*χ*^2^ = 0.19, *p* = 0.6642
	Rural	151 (52.8)	244 (54.3)	395 (53.7)	*χ*^2^ = 0.17, *p* = 0.6820
	Statistics and *p*-value	*χ*^2^ = 12.77, *p* = 0.0004	*χ*^2^ = 16.45, *p* < 0.0001	*χ*^2^ = 28.87, *p* < 0.0001	
Position	East	296 (63.1)	342 (63.9)	638 (63.6)	*χ*^2^ = 0.07, *p* = 0.7896
	Middle	142 (57.7)	210 (59.8)	352 (59.0)	*χ*^2^ = 0.27, *p* = 0.6067
	West	120 (61.5)	138 (58.7)	258 (60.0)	*χ*^2^ = 0.35, *p* = 0.5530
	Statistics and *p*-value	*χ*^2^ = 1.98, *p* = 0.3714	*χ*^2^ = 2.51, *p* = 0.2854	*χ*^2^ = 3.80, *p* = 0.1493	
Education attainment	Primary school or below	154 (52.4)	404 (58.8)	558 (56.9)	*χ*^2^ = 3.47, *p* = 0.0626
	Middle school	378 (66.0)	277 (66.3)	655 (66.1)	*χ*^2^ = 0.01, *p* = 0.9217
	College or higher	26 (60.5)	9 (56.3)	35 (59.3)	*χ*^2^ = 0.09, *p* = 0.7695
	Statistics and *p*-value	*χ*^2^ = 15.14, *p* = 0.0005	*χ*^2^ = 6.31, *p* = 0.0427	*χ*^2^ =17.78, *p* < 0.0001	
Physical activity	Low	46 (63.0)	45 (52.9)	91 (57.6)	*χ*^2^ = 1.63, *p* = 0.2015
	Moderate	242 (59.3)	198 (61.1)	440 (60.1)	*χ*^2^ = 0.24, *p* = 0.6218
	High	270 (62.9)	447 (62.8)	717 (62.8)	*χ*^2^ = 0.00, *p* = 0.9578
	Statistics and *p*-value	*χ*^2^ = 1.25, *p* = 0.5343	*χ*^2^ = 3.14, *p* = 0.2076	*χ*^2^ = 2.48, *p* = 0.2899	
Current smoking status	No	336 (62.0)	667 (61.9)	1003 (61.9)	*χ*^2^ = 0.00, *p* = 0.9629
	Yes	222 (60.3)	23 (53.5)	245 (59.6)	*χ*^2^ = 0.75, *p* = 0.3872
	Statistics and *p*-value	*χ*^2^ = 0.26, *p* = 0.6124	*χ*^2^ = 1.23, *p* = 0.2677	*χ*^2^ = 0.73, *p* = 0.3916	
Current drinking status	No	282 (63.4)	603 (61.3)	885 (61.9)	*χ*^2^ = 0.57, *p* = 0.4511
	Yes	276 (59.4)	87 (63.5)	363 (60.3)	*χ*^2^ = 0.76, *p* = 0.3831
	Statistics and *p*-value	*χ*^2^ = 1.55, *p* = 0.2137	*χ*^2^ = 0.25, *p* = 0.6163	*χ*^2^ = 0.48, *p* = 0.4900	
BMI categories	Normal weight	150 (61.7)	228 (63.5)	378 (62.8)	*χ*^2^ = 0.20, *p* = 0.6573
	Underweight	3 (30.0)	9 (50.0)	12 (42.9)	*χ*^2^ = 1.05, *p* = 0.3055
	Overweight	285 (65.2)	283 (61.4)	568 (63.3)	*χ*^2^ = 1.42, *p* = 0.2342
	Obese	120 (54.6)	170 (60.1)	290 (57.7)	*χ*^2^ = 1.55, *p* = 0.2135
	Statistics and *p*-value	*χ*^2^ = 11.21, *p* = 0.0107	*χ*^2^ = 1.86, *p* = 0.6011	*χ*^2^ = 8.83, *p* = 0.0316	

**Table 3 nutrients-15-00056-t003:** Intake of food groups by participants analyzed by quartile levels of dietary-pattern scores in this study (M (P_25_, P_75_), g/day).

Gender	Food Groups	Quartiles of Dietary Pattern Score				Total	Statistics and *p*-Value between Different Quartiles of Dietary Pattern Score
		*Q1*	*Q2*	*Q3*	*Q4*		
Male	Rice and its products	100.0 (50.0, 200.0)	150.0 (88.0, 200.0)	150.0 (60.0, 200.0)	100.0 (28.6, 200.0)	115.0 (50.0, 200.0)	*χ*^2^ = 16.59, *p* = 0.0009
	Wheat and its products	50.0 (14.3, 100.0)	50.0 (14.3, 100.0)	62.1 (19.8, 150.0)	150.0 (100.0, 250.0)	71.4 (21.4, 150.0)	*χ*^2^ = 137.73, *p* < 0.0001
	Corn and its products	25.7 (1.0, 60.0)	6.6 (0.0, 28.6)	4.0 (0.0, 17.9)	4.6 (0.0, 25.0)	7.1 (0.0, 34.3)	*χ*^2^ = 56.33, *p* < 0.0001
	Beans	1.7 (0.0, 8.6)	0.3 (0.0, 3.3)	0.2 (0.0, 2.6)	0.0 (0.0, 1.7)	0.3 (0.0, 3.3)	*χ*^2^ = 43.24, *p* < 0.0001
	Tuber crop	14.3 (2.7, 42.9)	14.3 (2.0, 42.9)	7.1 (0.0, 28.6)	14.3 (2.1, 57.1)	14.3 (1.6, 42.9)	*χ*^2^ = 10.49, *p* = 0.0149
	Fried dough foods	0.0 (0.0, 4.9)	0.0 (0.0, 3.3)	0.3 (0.0, 6.9)	2.1 (0.0, 18.1)	0.4 (0.0, 7.1)	*χ*^2^ = 34.59, *p* < 0.0001
	Soybean and its products	16.5 (6.0, 36.4)	12.8 (4.7, 26.9)	10.4 (4.0, 18.7)	8.3 (2.8, 21.6)	11.9 (4.0, 25.9)	*χ*^2^ = 30.67, *p* < 0.0001
	Vegetable	450.0 (250.0, 600.0)	400.0 (212.9, 503.3)	240.0 (160.0, 400.2)	202.4 (101.0, 323.3)	300.0 (200.0, 500.0)	*χ*^2^ = 120.50, *p* < 0.0001
	Algae and mushroom	18.6 (6.3, 37.1)	9.9 (3.5, 25.8)	8.3 (2.9, 17.6)	8.6 (2.6, 19.9)	10.1 (3.3, 25.7)	*χ*^2^ = 43.16, *p* < 0.0001
	Fruits	110.0 (38.6, 220.0)	62.9 (14.3, 150.0)	32.1 (4.9, 100.0)	16.0 (0.4, 77.1)	50.0 (7.9, 150.0)	*χ*^2^ = 105.02, *p* < 0.0001
	Dairy products	69.1 (0.0, 227.0)	26.8 (0.0, 187.0)	0.0 (0.0, 96.8)	0.0 (0.0, 106.4)	12.7 (0.0, 187.0)	*χ*^2^ = 37.88, *p* = 0.0001
	Meat	48.9 (20.9, 100.0)	43.2 (21.4, 80.8)	40.9 (20.0, 78.6)	40.0 (14.6, 67.6)	42.9 (18.8, 80.3)	*χ*^2^ = 6.22, *p* = 0.1013
	Processed meat	0.0 (0.0, 0.5)	0.0 (0.0, 0.7)	0.0 (0.0, 0.2)	0.0 (0.0, 0.0)	0.0 (0.0, 0.3)	*χ*^2^ = 4.61, *p* = 0.2026
	Poultry	9.9 (0.8, 21.4)	4.9 (1.1, 14.3)	3.3 (0.3, 8.2)	1.1 (0.0, 4.9)	3.3 (0.0, 13.2)	*χ*^2^ = 70.19, *p* < 0.0001
	Internal organs	0.0 (0.0, 0.0)	0.0 (0.0, 0.3)	0.0 (0.0, 0.4)	0.0 (0.0, 0.6)	0.0 (0.0, 0.4)	*χ*^2^ = 4.51, *p* = 0.2117
	Marine fish	2.1 (0.0, 14.3)	0.6 (0.0, 6.6)	0.0 (0.0, 3.3)	0.0 (0.0, 3.0)	0.3 (0.0, 6.6)	*χ*^2^ = 37.34, *p* < 0.0001
	Freshwater fish	14.3 (3.3, 42.9)	8.2 (1.4, 21.4)	4.9 (0.3, 14.3)	1.7 (0.0, 6.6)	6.6 (0.3, 16.5)	*χ*^2^ = 111.02, *p* < 0.0001
	Shrimp and crab	2.4 (0.0, 12.5)	1.1 (0.0, 7.7)	0.1 (0.0, 2.5)	0.0 (0.0, 1.4)	0.4 (0.0, 4.5)	*χ*^2^ = 67.79, *p* < 0.0001
	Mollusks	0.0 (0.0, 0.0)	0.0 (0.0, 0.0)	0.0 (0.0, 0.0)	0.0 (0.0, 0.0)	0.0 (0.0, 0.0)	*χ*^2^ = 2.66, *p* = 0.4478
	Eggs (except pidan)	45.5 (17.1, 56.7)	34.3 (15.7, 54.9)	28.9 (12.1, 55.3)	32.9 (15.7, 55.0)	34.3 (14.6, 55.0)	*χ*^2^ = 4.12, *p* = 0.2485
	Snacks and nuts	15.6 (1.7, 40.0)	7.1 (0.6, 24.3)	3.1 (0.0, 14.8)	2.3 (0.0, 18.2)	6.6 (0.0, 24.3)	*χ*^2^ = 45.02, *p* < 0.0001
	Beverages	0.0 (0.0, 0.0)	0.0 (0.0, 0.0)	0.0 (0.0, 0.0)	0.0 (0.0, 0.0)	0.0 (0.0, 0.0)	*χ*^2^ = 4.21, *p* = 0.2394
	Juice	0.0 (0.0, 0.0)	0.0 (0.0, 0.0)	0.0 (0.0, 0.0)	0.0 (0.0, 0.0)	0.0 (0.0, 0.0)	*χ*^2^ = 6.30, *p* = 0.0979
	Coffee	0.0 (0.0, 0.0)	0.0 (0.0, 0.0)	0.0 (0.0, 0.0)	0.0 (0.0, 0.0)	0.0 (0.0, 0.0)	*χ*^2^ = 5.56, *p* = 0.1351
Female	Rice and its products	100.0 (42.9, 150.0)	100.0 (50.0, 150.0)	150.0 (90.0, 200.0)	150.0 (50.0, 200.0)	100.0 (50.0, 200.0)	*χ*^2^ = 44.80, *p* < 0.0001
	Wheat and its products	50.0 (14.3, 100.0)	60.0 (21.4, 110.0)	50.0 (8.2, 100.0)	71.4 (14.3, 160.0)	50.0 (14.3, 107.1)	*χ*^2^ = 13.95, *p* = 0.0030
	Corn and its products	22.9 (4.0, 71.4)	10.0 (0.0, 30.0)	1.7 (0.0, 14.2)	1.7 (0.0, 11.2)	6.6 (0.0, 28.6)	*χ*^2^ = 123.17, *p* < 0.0001
	Beans	1.7 (0.0, 11.4)	0.6 (0.0, 4.3)	0.0 (0.0, 2.9)	0.0 (0.0, 1.7)	0.3 (0.0, 3.3)	*χ*^2^ = 62.30, *p* < 0.0001
	Tuber crop	14.3 (3.3, 57.1)	9.9 (1.2, 30.0)	8.6 (1.5, 28.6)	4.9 (0.0, 17.1)	8.6 (1.1, 28.6)	*χ*^2^ = 46.57, *p* < 0.0001
	Fried dough foods	0.0 (0.0, 2.1)	0.0 (0.0, 3.3)	0.0 (0.0, 2.3)	0.2 (0.0, 8.6)	0.0 (0.0, 3.3)	*χ*^2^ = 26.68, *p* < 0.0001
	Soybean and its products	11.6 (4.0, 26.9)	8.8 (3.7, 17.2)	7.1 (1.9, 14.5)	6.2 (1.8, 16.9)	8.2 (2.6, 19.2)	*χ*^2^ = 22.90, *p* < 0.0001
	Vegetable	303.3 (200.0, 505.3)	300.0 (200.0, 472.9)	217.1 (127.0, 400.0)	224.2 (121.4, 400.0)	276.6 (154.3, 450.0)	*χ*^2^ = 33.42, *p* < 0.0001
	Algae and mushroom	15.2 (4.4, 35.7)	7.5 (2.6, 21.4)	5.3 (1.0, 13.2)	3.8 (0.7, 10.7)	7.1 (1.6, 19.1)	*χ*^2^ = 91.00, *p* < 0.0001
	Fruits	130.0 (50.0, 220.0)	64.3 (14.3, 150.0)	22.9 (3.3, 66.0)	14.3 (0.0, 42.9)	43.6 (7.1, 123.0)	*χ*^2^ = 217.72, *p* < 0.0001
	Dairy products	42.7 (0.0, 241.9)	8.0 (0.0, 123.9)	0.0 (0.0, 42.2)	0.0 (0.0, 16.5)	0.0 (0.0, 107.1)	*χ*^2^ = 73.66, *p* < 0.0001
	Meat	28.6 (14.3, 60.1)	21.4 (7.8, 50.9)	20.3 (7.9, 50.0)	20.0 (7.1, 50.0)	21.4 (8.6, 51.3)	*χ*^2^ = 22.85, *p* < 0.0001
	Processed meat	0.0 (0.0, 0.3)	0.0 (0.0, 0.5)	0.0 (0.0, 0.3)	0.0 (0.0, 0.0)	0.0 (0.0, 0.2)	*χ*^2^ = 8.60, *p* = 0.0351
	Poultry	7.1 (0.6, 17.1)	2.9 (0.0, 7.1)	1.7 (0.0, 4.0)	0.7 (0.0, 3.3)	1.7 (0.0, 6.6)	*χ*^2^ = 118.43, *p* < 0.0001
	Internal organs	0.0 (0.0, 0.0)	0.0 (0.0, 0.0)	0.0 (0.0, 0.0)	0.0 (0.0, 0.0)	0.0 (0.0, 0.0)	*χ*^2^ = 9.39, *p* = 0.0246
	Marine fish	1.2 (0.0, 10.0)	0.1 (0.0, 3.3)	0.0 (0.0, 1.7)	0.0 (0.0, 1.9)	0.0 (0.0, 3.3)	*χ*^2^ = 42.64, *p* < 0.0001
	Freshwater fish	9.9 (0.8, 28.6)	3.3 (0.2, 14.3)	1.8 (0.0, 7.1)	0.5 (0.0, 3.3)	2.6 (0.0, 13.2)	*χ*^2^ = 115.63, *p* < 0.0001
	Shrimp and crab	0.6 (0.0, 7.1)	0.3 (0.0, 3.3)	0.0 (0.0, 0.7)	0.0 (0.0, 1.5)	0.0 (0.0, 3.0)	*χ*^2^ = 57.20, *p* < 0.0001
	Mollusks	0.0 (0.0, 0.0)	0.0 (0.0, 0.0)	0.0 (0.0, 0.0)	0.0 (0.0, 0.0)	0.0 (0.0, 0.0)	*χ*^2^ = 9.46, *p* = 0.0237
	Eggs (except pidan)	31.4 (14.3, 55.0)	21.4 (8.6, 50.0)	18.2 (5.3, 42.9)	25.7 (7.9, 56.6)	23.6 (7.9, 50.0)	*χ*^2^ = 20.56, *p*= 0.0001
	Snacks and nuts	11.4 (0.0, 50.0)	7.1 (0.4, 24.3)	2.4 (0.0, 10.3)	1.3 (0.0, 6.6)	3.3 (0.0, 20.1)	*χ*^2^ = 76.48, *p* < 0.0001
	Beverages	0.0 (0.0, 0.0)	0.0 (0.0, 0.0)	0.0 (0.0, 0.0)	0.0 (0.0, 0.0)	0.0 (0.0, 0.0)	*χ*^2^ = 1.45, *p* = 0.6946
	Juice	0.0 (0.0, 0.0)	0.0 (0.0, 0.0)	0.0 (0.0, 0.0)	0.0 (0.0, 0.0)	0.0 (0.0, 0.0)	*χ*^2^ = 29.71, *p* < 0.0001
	Coffee	0.0 (0.0, 0.0)	0.0 (0.0, 0.0)	0.0 (0.0, 0.0)	0.0 (0.0, 0.0)	0.0 (0.0, 0.0)	*χ*^2^ = 25.37, *p* < 0.0001

**Table 4 nutrients-15-00056-t004:** Macronutrient intake and percentage of energy from macronutrient intake analyzed by quartile levels of dietary-pattern score in this study (M (*P_25_*, *P_75_*)).

Gender		Quartiles of Dietary Pattern Score				Total	Statistics and *p*-Value
		*Q1*	*Q2*	*Q3*	*Q4*		
**Male**	Energy intake, kcal/day	2831.9 (1943.5, 3563.7)	2358.7 (1792.0, 3299.1)	2490.0 (1915.0, 3264.6)	2843.9 (1988.0, 3702.9)	2587.2 (1938.5, 3484.5)	*χ^2^* = 11.75, *p* = 0.0083
	Carbohydrate intake, g/day	381.9 (243.3, 550.2)	361.0 (228.4, 496.4)	371.4 (236.9, 504.0)	427.1 (276.7, 638.3)	382.8 (247.3, 562.5)	*χ^2^* = 10.65, *p* = 0.0138
	Percentage of energy from carbohydrate intake, %	61.8 (49.2, 71.0)	61.4 (49.5,71.6)	62.5 (50.7, 71.3)	66.2 (54.7, 72.4)	63.5 (50.7, 71.7)	*χ^2^* = 7.03, *p* = 0.0711
	Protein intake, g/day	89.4 (61.1, 113.5)	69.2 (53.4, 90.2)	63.6 (46.2, 80.9)	76.8 (52.2, 101.6)	71.2 (53.1, 98.3)	*χ^2^*= 51.93, *p* < 0.0001
	Percentage of energy from protein intake, %	12.8 (10.8, 15.6)	11.0 (9.6, 13.4)	10.2 (8.7, 12.2)	11.3 (9.2, 13.1)	11.2 (9.3, 13.5)	*χ^2^*= 78.81, *p* < 0.0001
	Fat intake, g/day	77.7 (50.7, 105.0)	67.9 (50.9, 96.0)	73.8 (54.5, 97.0)	76.3 (56.8, 101.0)	74.5 (53.8, 100.1)	*χ^2^* = 3.41, *p* = 0.3324
	Percentage of energy from fat intake, %	26.4 (19.1, 37.1)	28.0 (19.1, 37.5)	27.4 (20.1, 39.2)	24.6 (18.8, 34.4)	26.2 (19.1, 37.1)	*χ^2^* = 3.78, *p* = 0.2865
**Female**	Energy intake, kcal/day	2264.6 (1720.0, 2845.8)	2049.2 (1606.6, 2644.6)	2230.4 (1679.7, 2740.6)	2474.9 (2036.9, 3253.6)	2254.3 (1737.7, 2866.1)	*χ^2^* = 39.20, *p* < 0.0001
	Carbohydrate intake, g/day	329.5 (226.8, 452.0)	335.2 (219.0, 414.2)	356.5 (238.8, 447.7)	398.1 (310.5, 546.1)	359.1 (236.4, 460.8)	*χ^2^* = 37.42, *p* < 0.0001
	Percentage of energy from carbohydrate intake, %	62.0 (50.2, 70.2)	64.2 (53.7, 72.4)	63.8 (56.2, 72.8)	66.9 (57.7, 74.1)	64.1 (54.3, 72.6)	*χ^2^* = 18.50, *p* = 0.0003
	Protein intake, g/day	66.7 (49.5, 92.5)	58.9 (42.6, 75.2)	55.8 (41.0, 72.5)	67.0 (52.5, 85.3)	62.0 (46.1, 81.7)	*χ^2^* = 49.85, *p* < 0.0001
	Percentage of energy from protein intake, %	12.0 (10.2, 14.8)	10.9 (9.2, 13.1)	10.0 (8.5, 12.4)	10.7 (8.9, 12.8)	10.9 (9.1, 13.3)	*χ^2^* = 58.26, *p* < 0.0001
	Fat intake, g/day	69.8 (48.9, 92.8)	61.5 (41.7, 87.0)	61.6 (46.8, 88.6)	65.0 (47.3, 90.0)	64.0 (46.7, 89.7)	*χ^2^* = 13.08, *p* = 0.0045
	Percentage of energy from fat intake, %	29.4 (21.1, 38.8)	27.5 (19.0, 37.5)	28.1 (19.5, 35.9)	25.4 (17.0, 33.5)	27.1 (19.0, 36.7)	*χ^2^* = 15.15, *p* = 0.0191

**Table 5 nutrients-15-00056-t005:** The proportion of participants from different dietary pattern-scores groups who reached the recommended intake level of macronutrients (*n*, %) according to the *Guideline for the Prevention and Treatment of Type 2 Diabetes Mellitus in China (2020 edition)* [27].

Gender	Energy from Macronutrient Intake Compared with Guideline Recommendation	Quartiles of Dietary Pattern Score												Total		
		*Q1*			*Q2*			*Q3*			*Q4*					
		Low	Moderate	High	Low	Moderate	High	Low	Moderate	High	Low	Moderate	High	Low	Moderate	High
**Male**	PECI compared with the guideline recommendation	63 (27.5)	73 (31.9)	93(40.6)	57 (25.3)	75 (33.3)	93 (41.3)	55 (23.9)	76 (33.0)	99 (43.0)	44 (19.5)	57 (25.2)	125 (55.3)	219 (24.1)	281 (30.9)	410 (45.1)
	PEPI compared with the guideline recommendation	166 (72.5)	51 (22.3)	12 (5.2)	196 (87.1)	25 (11.1)	4(1.8)	214 (93.0)	16 (7.0)	0 (0.0)	216 (95.6)	9 (4.0)	1 (0.4)	792 (87.0)	101 (11.1)	17 (1.9)
	PEFI compared with the guideline recommendation	67 (29.3)	72 (31.4)	90 (39.3)	64 (28.4)	59 (26.2)	102 (45.3)	56 (24.4)	76 (33.0)	98 (42.6)	71 (31.4)	75 (33.2)	80 (35.4)	258 (28.4)	282 (31.0)	370 (40.7)
**Female**	PECI compared with the guideline recommendation	69 (24.9)	104 (37.6)	104 (37.6)	51 (18.0)	97 (34.3)	135 (47.7)	38 (13.4)	109 (38.4)	137 (48.2)	42 (15.2)	83 (30.0)	152 (54.9)	200 (17.8)	393 (35.1)	528 (47.1)
	PEPI compared with the guideline recommendation	208 (75.1)	55 (19.9)	14 (5.1)	247 (87.3)	30 (10.6)	6 (2.1)	262 (92.3)	18 (6.3)	4 (1.4)	242 (87.4)	26 (9.4)	9 (3.3)	959 (85.6)	129 (11.5)	33 (2.9)
	PEFI compared with the guideline recommendation	64 (23.1)	82 (29.6)	131 (47.3)	79 (27.9)	83 (29.3)	121 (42.8)	74 (26.1)	85 (29.9)	125 (44.0)	95 (34.3)	91 (32.9)	91 (32.9)	312 (27.8)	341 (30.4)	468 (41.8)

PECI: Percentage of energy from carbohydrate intake; PEPI: Percentage of energy from protein intake; PEFI: Percentage of energy from fat intake.

**Table 6 nutrients-15-00056-t006:** Fiber intake analyzed by quartile levels of dietary pattern score in this study (*M* (*P_25_*, *P_75_*)).

Gender	Variables	Quartiles of Dietary Patterns				Total	Statics and *p*-Value
		*Q1*	*Q2*	*Q3*	*Q4*		
**Male**	Fiber intake, g/day	21.5 (14.3, 31.3)	15.9 (12.2, 20.6)	12.0 (9.1, 17.2)	14.7 (9.1, 21.4)	15.6 (10.6, 22.2)	*χ*^2^ = 104.98, *p* < 0.0001
	Fiber intake per 1000 Kcal, g	8.26 (5.5, 12.1)	6.3 (4.7, 8.8)	5.0 (3.6, 7.2)	5.5 (3.8, 7.1)	6.0 (4.3, 8.5)	*χ*^2^ = 113.31, *p* < 0.0001
	Percentage of participants that meet the Chinese T2D guideline, *n*(%)	42 (18.3)	12 (5.3)	2 (0.9)	6 (2.7)	62 (6.8)	*χ*^2^ = 67.66, *p* < 0.0001
**Female**	Fiber intake, g/day	18.0 (13.0, 26.7)	14.2 (9.9, 20.1)	12.0 (7.9, 16.8)	12.6 (9.1, 19.0)	14.0 (9.8, 20.4)	*χ*^2^ = 89.38, *p* < 0.0001
	Fiber intake per 1000 Kcal, g	8.5 (6.0, 11.5)	6.7 (5.0, 9.5)	5.4 (4.0, 7.1)	5.3 (3.8, 7.0)	6.2 (4.5, 9.0)	*χ*^2^ = 154.10, *p* < 0.0001
	Percentage of participants that meet the Chinese T2D guideline, *n*(%)	37 (13.4)	22 (7.8)	8 (2.8)	13 (4.7)	80 (7.1)	*χ*^2^ = 26.84, *p* < 0.0001

Chinese T2D guideline: *Guideline for the Prevention and Treatment of Type 2 Diabetes Mellitus in China (2020 edition)* [27].

**Table 7 nutrients-15-00056-t007:** Clinical and anthropometric indicators of participants according to the quartile of dietary pattern scores (*M* (*P_25_*, *P_75_*)).

	Male					Female				
	*Q1*	*Q2*	*Q3*	*Q4*	Statistics and *p*-Value	*Q1*	*Q2*	*Q3*	*Q4*	Statistics and *p*-Value
*N* (%)	229 (25.2)	225 (24.7)	230 (25.3)	226 (24.8)		277 (24.7)	283 (25.2)	284 (25.3)	277 (24.7)	
WC, cm	90.0 (85.1, 96.8)	90.1 (83.6, 95.5)	90.0 (84.5, 96.2)	92.3 (85.1, 98.2)	*χ*^2^ = 5.29, *p* = 0.1516	86.0 (79.7, 93.5)	86.7 (80.5, 93.2)	86.0 (80.0, 92.3)	86.0 (80.0, 92.5)	*χ*^2^ = 1.62, *p* = 0.6546
WHtR	0.54 (0.50, 0.58)	0.54 (0.51, 0.58)	0.54 (0.51, 0.58)	0.55 (0.51, 0.59)	*χ*^2^ = 3.88, *p* = 0.2746	0.56 (0.51, 0.60)	0.56 (0.53, 0.60)	0.56 (0.52, 0.59)	0.56 (0.52, 0.60)	*χ*^2^ = 2.23, *p* = 0.5264
BMI, kg/m^2^	25.55 (23.89, 27.73)	25.77 (23.51, 27.51)	25.49 (23.29, 27.69)	26.52 (23.95, 28.46)	*χ*^2^ = 6.33, *p* = 0.0966	25.59 (23.22, 28.46)	25.63 (23.51, 28.36)	25.29 (23.12, 28.03)	25.42 (22.89, 27.47)	*χ*^2^ = 3.26, *p* = 0.3538
FPG, mmol/L	7.37 (6.23, 8.80)	7.83 (6.17, 9.93)	7.78 (6.14, 10.29)	8.06 (6.77, 10.86)	*χ*^2^ = 16.59, *p* = 0.0009	7.30 (5.92, 8.95)	7.35 (6.09, 9.18)	7.73 (6.10, 10.50)	7.90 (6.40, 10.57)	*χ*^2^ = 16.69, *p* = 0.0008
HbA_1c_, %	6.1 (5.4, 7.1)	6.3 (5.5, 7.6)	6.5 (5.5, 7.7)	7.2 (5.9, 8.5)	*χ*^2^ = 31.33, *p* < 0.0001	6.2 (5.4, 7.3)	6.1 (5.5, 7.3)	6.7 (5.6, 8.0)	6.8 (5.8, 8.3)	*χ*^2^ = 31.90, *p* < 0.0001
TC, mmol/L	4.50 (3.90, 5.03)	4.65 (3.94, 5.29)	4.48 (3.86, 5.25)	4.75 (4.12, 5.54)	*χ*^2^ = 13.15, *p* = 0.0043	5.05 (4.35, 5.77)	4.91 (4.27, 5.53)	5.08 (4.36, 5.78)	5.15 (4.39, 5.91)	*χ*^2^ = 6.69, *p* = 0.0826
TG, mmol/L	1.46 (0.96, 2.18)	1.42 (1.01, 2.28)	1.41 (1.00, 2.18)	1.58 (1.10, 2.60)	*χ*^2^ = 5.74, *p* = 0.1251	1.52 (1.13, 2.35)	1.51 (1.09, 2.31)	1.55 (1.04, 2.34)	1.56 (1.06, 2.33)	*χ*^2^ = 1.47, *p* = 0.6890
LDL-C, mmol/L	2.78 (2.32, 3.29)	3.00 (2.37, 3.64)	2.86 (2.32, 3.41)	3.02 (2.51, 3.68)	*χ*^2^ = 11.37, *p* = 0.0099	3.25 (2.68, 3.88)	3.09 (2.46, 3.76)	3.19 (2.61, 3.87)	3.24 (2.61, 4.01)	*χ*^2^ = 6.82, *p* = 0.0779
HDL-C, mmol/L	1.04 (0.90, 1.24)	1.08 (0.91, 1.26)	1.05 (0.87, 1.26)	1.08 (0.92, 1.25)	*χ*^2^ = 0.93, *p* = 0.8185	1.19 (0.99, 1.37)	1.17 (0.97, 1.37)	1.19 (1.04, 1.45)	1.17 (1.00, 1.43)	*χ*^2^ = 3.83, *p* = 0.2799
Glycemic control (*n*, %)	167 (72.9)	144 (64.0)	144 (62.6)	103 (45.6)	*χ*^2^ = 37.47, *p* < 0.0001	198 (71.5)	191 (67.5)	156 (54.9)	145 (52.4)	*χ*^2^ = 30.94, *p* < 0.0001
Impaired fasting glucose (*n*, %)	138 (60.3)	142 (63.1)	145 (63.0)	163 (72.1)	*χ*^2^ = 7.94, *p* = 0.0472	163 (58.8)	163 (57.6)	181 (63.7)	189 (68.2)	*χ*^2^ = 8.43, *p* = 0.0379

## Data Availability

The data are not permitted to be disclosed according to the National Institute for Nutrition and Health, Chinese Center for Disease Control and Prevention.

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
