# Peer review of "Dietary Pattern Associated with the Risk of Poor Glycemic Control in Chinese Diabetic Adults: Results from the China Nutrition and Health Surveillance 2015–2017 Survey"

_nutrients, 2022, doi:10.3390/nu15010056_

Round 1

Reviewer 1 Report

The authors propose a very interesting report on the eating habits of the Chinese population:

The discussions are also well organized and adequately embody all aspects.

Conduct the Mediterranean diet correctly, but it should be emphasized that it should not necessarily be considered with the same foods (it would be impossible, given the considerable cultural and environmental difficulties between the two). Still, the peculiarities should be considered, i.e., foods rich in polyphenols and fibers, as shown, for example, in 10.3390 / antiox10020328. This is in line with the outcomes of the manuscript, as the higher incidence is correlated with lower consumption of vegetables.

Even if it is from a nutritional point of view, the incidence of physical activity should be better emphasized in terms of frequency, intensity, and type.

A point to consider better in future studies would be body composition via a DXA and/or bioimpedance vector analysis.

Author Response

Dear Professor:

Thank you very much for your comments and suggestions. Your suggestions have helped me a lot in the process of revising my manuscript.

Conduct the Mediterranean diet correctly, but it should be emphasized that it should not necessarily be considered with the same foods (it would be impossible, given the considerable cultural and environmental difficulties between the two). Still, the peculiarities should be considered, i.e., foods rich in polyphenols and fibers, as shown, for example, in 10.3390 / antiox10020328. This is in line with the outcomes of the manuscript, as the higher incidence is correlated with lower consumption of vegetables.

In response to your first issue, I have carefully read and cited the article you mentioned during the manuscript revision process, and I strongly agree with the benefits of the antioxidant and anti-inflammatory effects of polyphenolic compounds in diabetic patients. Your suggestion to measure the body composition of the investigated subjects by DXA and bioimpedance vector analysis is indeed a direction for our future research, because muscle weight is strongly associated with glucose metabolism in diabetic patients, and this index is more closely related to glycemic control in diabetic patients than body weight. This point is also illustrated in our revised article, i.e., to increase the measurement of body composition in diabetic patients in future studies.

Even if it is from a nutritional point of view, the incidence of physical activity should be better emphasized in terms of frequency, intensity, and type.

The second issue you mentioned, about physical activity, is also of great importance to us. In this study, we calculated the metabolic equivalent of energy (MET) to assess the amount of physical activity in diabetic patients and classified the intensity of physical activity as high, moderate, and low. However, it is difficult to distinguish the type of exercise and the corresponding intensity because in the questionnaire for this study, no distinction was made between physical activity such as resistance exercise, aerobic exercise, or combined exercise. Nonetheless, by calculating METs, we were able to indirectly describe the frequency and intensity of physical activity of the participants. In the present study, the METs of male, female and total participants were illustrated respectively in Table1, and also used as covariates in multiple-variable adjusted logistic regression. We also reviewed the literature and improved the exercise recommendations for diabetic patients in the section of discuss.

A point to consider better in future studies would be body composition via a DXA and/or bioimpedance vector analysis.

Special thanks to you for your good comments.

Reviewer 2 Report

The manuscript is well written, the methodology used is appropriate and sufficient. The results support the discussion. However, I do have some comments.

I. Major comments:

1. It would be great if the authors could include the intake of dietary fiber, refined carbohydrates and fructose. Because these nutrients or components of food influence glycemia.

2. I suggest detailing the intake of fatty acids, such as SFA, MUFA, PUFA, n-3 PUFA and n-6 PUFA.

3. I suggest briefly discussing how the dietary pattern in people with diabetes, in addition to being related to glycemic control, can influence the development of other chronic pathologies, such as NAFLD.

4. It would be interesting to include a figure that summarizes the main results of the study.

II. Minor comments:

1. Improve the wording of the objective of the study.

2. Figure 2.a. and Figure 2.b., are very small, it was difficult for me to read them.

Author Response

Dear Professor:

Thank you for your kind advice! Your suggestions have helped me a lot in the process of revising my manuscript.

1. It would be great if the authors could include the intake of dietary fiber, refined carbohydrates and fructose. Because these nutrients or components of food influence glycemia.

2. I suggest detailing the intake of fatty acids, such as SFA, MUFA, PUFA, n-3 PUFA and n-6 PUFA.

In response to your questions 1 and 2, first, as for the data on dietary fiber that you mentioned, it is strongly associated with glycemic control in diabetics. Following your suggestion, we completed the calculation of the results and calculated the percentage of dietary fiber intake that could meet the recommendations compared with Prevention and Treatment of Type 2 Diabetes Mellitus in China (2020 edition) and added it to the revised manuscript. However, I am so sorry to inform you that it is not appropriate to provide more information on fatty acid intake among the participants in the current study, since a member of our group has been working on the intake of fructose and fatty acids; he is independently developing a database on fructose and fatty acids, which has not been shared; and he is currently working on a manuscript on related research, so it is not a suitable time to report it in my manuscript. In view of your concern, I have pasted the title and abstract of the published literature in the attached document. I hope it will help you better understand the dietary fatty acid and fructose intake among Chinese people.

3. I suggest briefly discussing how the dietary pattern in people with diabetes, in addition to being related to glycemic control, can influence the development of other chronic pathologies, such as NAFLD.

In response to your question 3, I agree with you that HbA1c should not be the only management goal for diabetic patients, and good dietary patterns also play a positive role in other co-morbidities of diabetic patients, including improving symptoms of diabetes combined with non-alcoholic liver disease, cognition, and lipid metabolism, which I have discussed in the revised manuscript.

4. It would be interesting to include a figure that summarizes the main results of the study.

For the graphs summarizing the results, Figure 1 shows the factor loadings for all food groups and facilitates a better presentation of the differences between the individual factors. In addition, to ensure readability for the reader, we added “*” to food groups with absolute values of factor loadings greater than 0.25 in the figure to help the reader understand the characteristics of this dietary pattern more easily.

II. Minor comments:

1. Improve the wording of the objective of the study.

2. Figure 2.a. and Figure 2.b., are very small, it was difficult for me to read them.

Following your advice, I strongly agree with the role of polyunsaturated fatty acids in the Mediterranean diet, namely their anti-inflammatory, antioxidant, and lipid metabolism improving effects. In the further study, we will revise the questionnaire involving fat intake, investigate the types and amounts of edible oils in more detail, and create a more comprehensive database of fatty acids in Chinese edible oils and foods, to provide more evidence for the mechanistic exploration of dietary patterns.

Special thanks to you for your good comments.

Round 2

Reviewer 2 Report

Manuscript was improved. Authors answered all my comments.